# LncRNA-Mediated Adipogenesis in Different Adipocytes

**DOI:** 10.3390/ijms23137488

**Published:** 2022-07-05

**Authors:** Peiwen Zhang, Shuang Wu, Yuxu He, Xinrong Li, Yan Zhu, Xutao Lin, Lei Chen, Ye Zhao, Lili Niu, Shunhua Zhang, Xuewei Li, Li Zhu, Linyuan Shen

**Affiliations:** 1College of Animal Science and Technology, Sichuan Agricultural University, Chengdu 611130, China; zpw1995@stu.sicau.edu.cn (P.Z.); wushuang@stu.sicau.edu.cn (S.W.); heyuxu@stu.sicau.edu.cn (Y.H.); 2020302147@stu.sicau.edu.cn (X.L.); 202000576@stu.sicau.edu.cn (X.L.); chenlei815918@sicau.edu.cn (L.C.); zhye@sicau.edu.cn (Y.Z.); niulili@sicau.edu.cn (L.N.); 14081@sicau.edu.cn (S.Z.); xuewei.li@sicau.edu.cn (X.L.); 2Farm Animal Genetic Resources Exploration and Innovation Key Laboratory of Sichuan Province, Sichuan Agricultural University, Chengdu 611130, China; 3College of Life Science, China West Normal University, Nanchong 637009, China; zhuyan0720@stu.cwnu.cn

**Keywords:** long noncoding RNA, brown fat, ectopic fat, beige fat, adipose tissue

## Abstract

Long-chain noncoding RNAs (lncRNAs) are RNAs that do not code for proteins, widely present in eukaryotes. They regulate gene expression at multiple levels through different mechanisms at epigenetic, transcription, translation, and the maturation of mRNA transcripts or regulation of the chromatin structure, and compete with microRNAs for binding to endogenous RNA. Adipose tissue is a large and endocrine-rich functional tissue in mammals. Excessive accumulation of white adipose tissue in mammals can cause metabolic diseases. However, unlike white fat, brown and beige fats release energy as heat. In recent years, many lncRNAs associated with adipogenesis have been reported. The molecular mechanisms of how lncRNAs regulate adipogenesis are continually investigated. In this review, we discuss the classification of lncRNAs according to their transcriptional location. lncRNAs that participate in the adipogenesis of white or brown fats are also discussed. The function of lncRNAs as decoy molecules and RNA double-stranded complexes, among other functions, is also discussed.

## 1. Introduction

About only 1–2% of human DNA is transcribed into mRNAs that code for proteins, whereas the rest is coded into noncoding RNAs (ncRNAs) [1]. The noncoding RNAs include snRNA (small nuclear RNA), siRNA (small interfering RNA), miRNA (microRNA), lncRNA (long noncoding RNAs), and circRNA (circular RNA), among others [2]. lncRNAs are noncoding RNAs present in numerous eukaryotes, first reported in 1990. Since a few lncRNAs can code for proteins, the naming and definition of long noncoding RNAs are still controversial [3]. For a long time, lncRNAs have been considered redundant transcripts, commonly called “noise sequences” [4]. However, emerging evidence shows that lncRNAs have strong temporal and spatial expression specificity and tissue specificity [5], are poorly conserved among species [6,7], and regulate numerous biological processes [8]. lncRNAs regulate gene expression at multiple levels [9]. Advances in high-throughput sequencing and other biotechnology have revealed the molecular mechanisms by which lncRNA regulates biological functions [10]. At present, the functions of lncRNAs are yet to be exhausted, and much remains to be discovered.

Obesity is currently a public health concern globally. According to the World Health Organization (WHO), 1.9 billion adults (39%) are overweight (BMI > 25) and 600 million people (13%) are obese (BMI > 30). Obesity metabolic syndrome is caused by excessive energy intake, in which the triglycerides that accumulate in the body are transformed into fat [11,12,13]. Obesity is a chronic recurrent disease characterized by excessive accumulation of fats in the body. The pathogenesis of obesity is quite complex, and many factors, including genetics, viral infections, insulin resistance, inflammation, gut microorganisms, and abnormal hormone secretion are implicated in obesity development. Chronic obesity causes systemic metabolic diseases, high blood pressure, hip cartilage, and other bone and joint diseases. Accumulation of large amounts of visceral fat can cause insulin resistance, fatty liver disease, type 2 diabetes, liver fibrosis, liver cancer, and other diseases. Similarly, high body fat can cause hypertriglyceridemia and irreversible chronic diseases such as atherosclerosis. At the same time, obesity predisposes childbearing age women to polycystic ovary syndrome or gestational diabetes mellitus, increases the maternal metabolic burden, and has irreversible effects on methylation of maternally imprinted genes in offspring. In addition, obesity appears to increase the severity of Corona Virus Disease 2019, (COVID-19) [14,15]. Obesity and vitamin deficiency are major risk factors for poor prognosis after COVID-19 infection. In addition, obesity, diabetes, and hypertension increase the risk of COVID-19 infection and severe COVID-19 [15].

Adipose tissue is an important metabolic and endocrine site and the body’s most important energy storage site [16,17]. Therefore, many scholars have gradually shifted their attention to the molecular mechanism of how lncRNAs regulate fat secretion and metabolism, since lncRNAs regulate numerous biological processes, including fat secretion, lipid metabolism, etc. This article reviews the literature with regard to the role of lncRNAs on the development and function of adipocytes and provides references for subsequent research.

## 2. LncRNA

lncRNAs are small RNA transcripts greater than 200 nt. lncRNAs can be divided into seven types based on the transcription position from the genome (Figure 1). Intergenic lncRNAs (intergenic lncRNA), also called lincRNAs, are intergenic region lncRNAs, transcribed from the middle region of two coding genes and at least 1 kb away from the coding gene. They mainly regulate cellular activities. Intron lncRNAs (intronic lncRNAs), mainly produced in the intron region of the coding gene, have corresponding coding genes and the same expression pattern, and they primarily regulate the expression of genes. Antisense lncRNA (antisense lncRNA), mainly produced in the antisense strand of the coding strand, binds and regulates mRNAs’ expression. Sense lncRNAs (sense-overlapping lncRNA) are transcribed in a direction similar to that of adjacent mRNA and partial or complete overlap exons. They contain an open reading frame (ORF) for protein translation but do not code for protein, restricting the mRNA translation process by mediating the stop codon; promoter-associated lncRNA and untranslated-region overlapping lncRNA (untranslated-region overlapping lncRNA) bind to the promoter and untranslated region of the regulated mRNA. Enhancer sub-type lncRNA (enhancer lncRNA) mainly regulates the expression of neighboring genes through binding cis-regulatory sites or enhancers [18,19,20,21].

Like mRNA-encoding protein, lncRNA has a 5’cap and a 3’poly A tail. lncRNAs use the same gene as a transcription template to form different lncRNA transcripts by variable shearing. Unlike mRNA, lncRNA has strong tissue specificity, and its abundance is lower than that of mRNA. lncRNA exists in the nucleus, cytoplasm, and organelles, but they are more abundant in the nucleus than in the cytoplasm and organelles [22,23,24]. lncRNA regulates proliferation, differentiation, apoptosis of cells, and the development of tissues and organs. For example, lncLSTR (liver-specific triglyceride regulator RNA) is an lncRNA specifically expressed in mouse liver and regulates energy metabolism and lipid metabolism in the liver by directly regulating one of the rate-limiting enzymes responsible for bile acid synthesis, Cyp8b1 [25].

## 3. The Mechanism of Action of LncRNA

LncRNA regulates gene expression through epigenetic modification at transcription and post-transcriptional levels. Accordingly, lncRNA regulates numerous biological processes, including proliferation, differentiation, apoptosis of cells, and carcinogenesis.

Most lncRNAs are transcribed and mature in the nucleus. About one-third of lncRNAs are exclusively expressed in the nucleus, and they regulate gene expression through epigenetic modification and at the transcription level [26] (Figure 2). lncRNAs mainly regulate gene expression by promoting the binding of epigenetic proteins to DNA [26]. In the cytoplasm, lncRNA regulates the decay of mRNA or sponges microRNA at the post-transcriptional level and, thus, protects target mRNA from microRNA inhibition [27]. For example, Xist, the first lncRNA discovered, can attach to the entire X chromosome, disrupting the methylation of numerous histones on the chromosome [28]. Additionally, by interacting with laminB receptors, it also changes the three-dimensional structure of DNA in the nucleus. Finally, Xist can pervade and silence the entire X chromosome [29]. Antisense lncRNA Khps1 recruits histone acetyltransferase p300/CBP to the SPHK1 promoter, which changes the chromatin structure and allows E2F1 to bind and transcriptionally activate SPHK1. This promotes the progression of the cell cycle, but apoptosis and tumorigenesis are regulated [30]. In eukaryotes, lncRNA regulates the initiation, extension, and termination of transcription of neighboring and distant genes in cis- or transpositions. lncRNA upstream of the protein-encoding gene disrupts transcription factor binding to the promoter region of the downstream gene, inhibiting transcription [31]. lncRNA can regulate the expression of distant genes by binding gene promoters, transcription factors, and Po1Ⅱ [32,33]. In addition, the subnuclear structure, including nucleoli, nuclear speckles, and side spots, also plays a vital role in gene expression regulation [34]. lncRNA also regulates gene expression by disrupting the assembly, maintenance, and function of the subnuclear structure [35,36]. For example, mRNAs are mainly stored in the paraplaque. lncNEAT1 assembles paraplaque structure by recruiting proteins that inhibit gene expression [37].

lncRNA also regulates gene expression at the post-transcriptional level. lncRNAs also interfere with post-translational modification of proteins, causing abnormal signal transduction [38]. lncRNA NKILA disrupts the phosphorylation of IκB and inhibits NF-κB and nuclear factor kappa-B (NF-κB) activation, modulating breast cancer metastasis [39]. In addition, lncRNA affects the translation and stability of mRNA precursors. For example, lncMALAT1 indirectly regulates the selective splicing of intracellular mRNA by regulating the distribution and phosphorylation of the serine/arginine (SR)-splicing factor in the nucleus and cytoplasm [40]. lncRNA GAS5 binds to the translation initiation factor eIF4E and c-Myc mRNA, inhibiting c-Myc protein expression [41]. Antisense lncRNA forms an RNA-RNA dimer with the mRNA transcribed from the neighboring genes. This increases the stability of mRNA for these genes and, thus, upregulates the expression of the respective genes. The lncRNA BACE1-AS-BACE1 mRNA dimer modulates the effect of RNase on BACE1 mRNA degradation, thereby increasing the expression of the BACE1 gene [42]. The mechanism through which lncRNAs perform their function as competitive endogenous RNA (ceRNA) remains a matter of research interest. Wu [43] recently reported that lncSNHG15 binds miR-200-3p through a sponge mechanism, upregulating the expression of the YAP1 gene. This promotes the tumorigenesis of papillary thyroid cancer. Sirtl AS lncRNA regulates muscle development by competing with miR-34a for binding Sirtl mRNA [44]. The lncRNA can also act as a molecular sponge competitively adsorbing proteins to regulate the transcription and translation of mRNAs, thereby regulating the progression of various biological processes. For example, Zhang [45] et al. showed that LncRNA MACC1-AS1 stabilized the stability of the lncMACC1-AS structure by binding to the RNA-binding protein PTBP1, which also enhanced its adsorption to miRNAs. In addition, lncRNAs regulate protein activity and other biological functions by acting as miRNA precursors [46].

## 4. Types and Functions of Adipose Tissue

Research has revealed that fat is a tissue and an organ that secretes important biological molecules. The adipose tissue can be divided into white adipose tissue (WAT), brown adipose tissue (BAT) [47], beige adipose tissue [48], and ectopic lipid deposition (ELD) [49,50]. Brown adipose tissue is usually UCP1^+^ cells, while white adipose tissue cells contain UCP1^−^ white adipocytes as well as UCP1^+^ beige adipocytes. Beige adipocytes may be derived from WAT precursor cells or may be transformed from mature white adipocytes after stimulation or gene regulation [51]. It is important to note that ectopic fat is not a type of adipose tissue, but rather a biological process of abnormal fat deposition [52,53]. Several important biological molecules, including hormones, regulatory factors, and exosomes, which can act on the same or other tissues and organs to regulate metabolism and development, are secreted in the adipose tissue [54]. Among them, fat-derived exosomes can remotely regulate the growth and development of tissues and organs via miRNA, lncRNA, and other factors [55]. Chen [56] reported that exosomes derived from brown fat are coated with many miRNAs, among which miR-92a, detectable in blood, regulates the activity of brown fat. Therefore, miR-92a in the exosomes is a marker for the brown fat activity mark. Adipose tissue is derived from the mesoderm and is divided into paraxial and lateral mesoderm. White adipocytes originate from the two mesoderms, and only the paraxial mesoderm can produce brown adipocytes. Even though the two adipocytes originate from the same source, their development mode and structure are distinct.

The appearance and development of white adipose tissue are later than brown fat. White fat differentiation begins immediately after birth, whereas brown fat differentiation begins at the embryonic stage. At the embryonic development stage, brown fat appears earlier than white fat [51,57]. In addition, the morphology and biological functions performed by the two are also different. White adipocytes are single lipid droplets with a larger cell area and fewer blood vessels. White adipose tissue is stored under the skin and around the organs and stores excess energy in the body in the form of triglycerides [58,59].

Brown adipocytes are relatively fewer in the body and are distributed differently across species and age groups of the same species. In the human body, brown adipocytes are mainly distributed around the kidney, neck, supraclavicular, and pericardium [60]. Moreover, the amount of these fats in the body decreases with age. Brown fat cells have multicompartment lipid droplets, small cell areas, numerous blood vessels, and dense mitochondria. Therefore, brown fat can oxidate phosphorylate mitochondria via many uncoupling proteins (UCPl) on the surface of the mitochondrial inner membrane. Uncoupling promotes free fatty acid metabolism. UCP1 in the inner mitochondrial membrane mediates exothermic respiration. Although the proton gradient generated by the electron transport chain is used for mitochondrial adenosine triphosphate (ATP) synthesis, UCP1 generates heat using the proton gradient instead of ATP synthesis and consumption energy in the form of heat [60,61]. Brown fat participates in thermogenesis and non-tremor thermogenesis (NST). A recent report showed that the thermogenesis of activated brown adipose tissue uses about 4.5% of human energy [62].

In recent years, the thermogenesis mechanism of brown fat has attracted the interest of researchers, particularly its role in preventing obesity. Many studies have shown that under certain conditions, white and brown fat are converted into each other [63]. Moreover, under cold stimulation, some white adipose tissue exhibits a brown fat phenotype and displays the thermogenic activity of these fats [64]. Brown white fat is called “beige fat” [65]. The browning of white adipose tissue mainly results from sympathetic stimulation and the interaction of norepinephrine (NE) and β3-adrenergic receptors (β3-ARs) on the cell membrane of white fat cells. In contrast, cold can stimulate an exothermic reaction in white fat. Therefore, cold is one of the factors that can strongly stimulate the sympathetic nerves [66]. Nuria [67] found that inhibiting the expression of MKK6 in white fat promotes the browning of white fat by increasing the oxidative phosphorylation of UCP1. Studies have also shown that specific stimuli such as cold and adrenaline can induce thermogenesis in inguinal white adipose tissue (iWAT) by upregulating the expression of UCP1 [68]. Studies have shown that beige and brown adipocytes have similar heat-producing abilities. However, the mechanisms underlying brown fat proliferation and differentiation and browning of white fat overlap, but are regulated by different genes [52].

Ectopic deposition of fat is another form of fat storage. Under normal conditions, there is a balance in the oxidative metabolism of free fatty acids (FFA) in various body tissues. However, when FFA exceeds the body’s oxidative capacity, FFAs are transferred from the adipose tissue to nonfat tissues, such as skeletal muscle, myocardium, liver, and pancreas [69,70,71], etc. FFA is esterified in these nonfat tissues and stored in the cytoplasm in the form of triglycerides. For the animal husbandry industry, proper lipid deposition in muscles can improve the taste and flavor of meat products. For humans, excessive ectopic fat deposition can cause obesity and related metabolic syndromes such as heavy heart, nonalcoholic fatty liver disease, and insulin resistance, among others [71,72].

## 5. LncRNA and Adipose Tissue

High-throughput sequencing technology has revealed many adipogenesis and development-related lncRNAs (Appendix A). However, many regulatory pathways related to obesity and lipid deposition have not been fully uncovered. Studies have shown that excessive lipid accumulation in adipose tissue is associated with dysregulated lncRNA expression. The amount of fats also affects the quality of livestock and poultry meat products [73]. Therefore, clarifying how lncRNA participates in fat development can further research in fat development-related diseases and the production of quality meat products. The biological functions of white, brown, and beige fats and lncRNAs related to this process are discussed (Figure 3, Figure 4 and Figure 5). As shown in Figure 3, Figure 4 and Figure 5, lncRNAs are regulated in a variety of ways in adipocytes to regulate lipid deposition or to regulate white fat browning. The most commonly reported mode of regulation is that of ceRNA regulation, where lncRNAs regulate the transcription or translation of target genes by competitively binding miRNAs or proteins, such as lncGAS5 [41]. lncRNAs also regulate lipogenic differentiation by acting as decoy molecules, RNA-RNA dimers, histone modifications, regulation of target-gene-promoter transcriptional activity and signaling pathway activity, and white fat browning, among other processes. In addition, it is worth noting that adipose-derived MSCs, as a class of pluripotent stem cells, are an excellent cell model for studying the biological processes of lipogenic differentiation, but most studies have been conducted on mature adipocytes. However, most of the studies are based on mature adipocytes. There are very few studies on adipose-derived MSCs and adipose-progenitor-cell-related lncRNAs and the regulatory mechanisms are not well studied, such as how lncRNAs are involved in the transduction of signaling pathways and the regulatory relationship with the host gene. In this paper, we summarize seven lncRNAs that regulate adipose-derived MSCs, such as TINCR [74], AC092834.1 [75], lnc13728 [76], and others.

## 6. Different Regulatory Modes of LncRNAs in Adipogenic Processes

### 6.1. Decoy Molecules

Bai et al. [77] found that lncBATE10 is overexpressed in brown adipose tissue. The lncRNA is a 1.4 Kb molecule measuring 22 and contains four exons of intergenic lncRNA. Norepinephrine and adenosine cyclophosphate (cAMP) upregulate the expression of lncBATE10 in brown adipocytes. cAMP regulates the expression of genes downstream of brown fat markers such as Ucp1 and Pgc1α by regulating the phosphorylation and activation of the transcription factor Creb. Moreover, the specific Creb-binding site is in the lncBATE10 promoter region. The luciferase report and experiment has shown that Creb regulates the activity of promoter lncBATE10, indicating that lnc-BATE10 is regulated in the cAMP-Creb regulatory axis. RNA-pulldown and RIP experiments have shown that Celf1, a tan RNA-binding protein, binds to lncBATE10. Further experiments have shown that lncBATE10 promotes the differentiation of brown adipocytes, and the effect of fat browning by competitively binding to Celf1 to induce Pgc1a release.

Similar to lncBATE10, the AU-rich elements (AREs) in CAAlnc1 are necessary for the interaction between cachexia-related anti-adipogenesis lncRNA 1 (CAAlnc1) and Hu antigen R (HuR), and inhibiting these elements disrupts the binding to HuR. The interaction between CAAlnc1 and HuR mainly blocks the binding of HuR to CCAAT/enhancer-binding protein α (C/EBPα) and peroxisome-proliferator-activated receptor γ (PPARγ). The latter are the two main adipogenic transcription factors (TFs) involved in the final adipogenesis process. In general, therefore, CAAlnc1 inhibits adipogenesis by blocking terminal differentiation [78,79]. Meanwhile, lncXIST [80] inhibits lipid deposition by binding to C/EBPα to perform the function of decoy molecules.

### 6.2. RNA-RNA Dimer

Adiponectin (AdipoQ) is a hormone secreted by fat cells. It mainly regulates insulin sensitivity and glucose metabolism. AdipoQ AS lncRNA is an lncRNA transcribed from the antisense strand of the adiponectin 3 chromosome [81]. Studies have shown that AdipoQ AS lncRNA is strongly expressed in mouse gonadal fat, and its expression is upregulated during the differentiation of 3T3-L1 cells and primary preadipocytes. Since the fluorescence intensity is much stronger in the cytoplasm than that in the nucleus of mature adipocytes, it is thought that most AdipoQ lncRNA is translocated to the cytoplasm to perform its function. AdipoQ AS lncRNA inhibits the cytoplasmic translation of AdipoQ mRNA and the subsequent fat formation by forming a double strand with AdipoQ mRNA. In addition, the nonoverlapping regions of AdipoQ AS lncRNA and AdipoQ AS lncRNA inhibit fat formation through double-strand formation. AdipoQ AS lncRNA improves the blood glucose tolerance and insulin sensitivity of HFD mice and upregulates the expression of thermogenesis-related genes, including UCP1, PGC1-α and PRDM16 levels. Additionally, compared with AdipoQ mRNA, AdipoQ AS lncRNA more strongly participates in lipid metabolism. PU.1 AS is another antisense lncRNA that participates in this regulation process. In particular, it binds and inhibits PU.1 mRNA expression and promotes adipogenic differentiation [82].

Unlike AdipoQ AS, Zhao et al. [83] found that Blnc1, mainly found in the nucleus, is overexpressed during the differentiation of brown fat and in cold environments. Thus, Blnc1 alone cannot activate the expression of thermogenic genes but performs this process in combination with hnRNPU to play a role. By recruiting early B cell factor 2 (EBF2) to form a ribonucleoprotein complex, it enhances the transcriptional activity of Blnc1, and EBF2 will also be recruited to the proximal promoter of Blnc1 and stimulate its expression and play a role in inducing UCP1 and mitochondrial gene expression. Therefore, the Blnc1 gene is the direct EBF2 target. Blnc1 promotes the expression of the thermal gene during the formation of brown and beige fat by forming a transcription complex with EBF2. This generates a positive feedback loop that promotes the differentiation of brown and beige fat cells. Moreover, the lncRAP2-Igf2bp2 [84] complex enhances adipogenesis and energy use, which are linked to obesity-related diabetes.

### 6.3. MicroRNA Sponge

lncADNCR is the most downregulated lncRNA among the differentially expressed lncRNAs in bovine preadipocytes and differentiated adipocytes. The lncADNCR is nearly without coding capacity, and the lncRNA is mainly distributed in the cytoplasm of preadipocytes. In recent years, increasing evidence has shown that lncRNA can regulate genes by directly competing for mRNA binding with miRNAs, similar to endogenous RNAs (ceRNAs) [85]. ADNCR inhibits adipocyte differentiation by increasing the expression of Sirtuin type 1 (SIRT1) gene, the miR-204 target, by competing with the miRNA. SIRT1 inhibits adipocyte differentiation and adipogenic gene expression by combining with NCoR and SMART, disrupting the activity of PPARγ. Therefore, ADNCR blocks miR-204 function, increasing the expression of miR-204 target gene SIRT1 and, thus, inhibiting adipocyte differentiation [86]. However, given the poor lncRNA conservation, the ADNCR expression has only been assessed in mice but not in humans. Further experiments are needed to ascertain whether ADNCR exists in humans.

By contrast, Gm15290 is overexpressed in white fat, promotes adipogenesis, and participates in obesity development. miR-27b directly targets Gm15290. Additionally, miR-27b targets PPARγ during the adipogenic differentiation stage [87]. Gm15290 is thought to activate PPARγ through ceRNA as miR-27b, promoting adipogenesis and fat deposition [87]. Moreover, lncRNAs in Table 1 also adsorb their targeted microRNAs through the sponge mechanism, regulating the adipogenic differentiation of cells. In addition, as we mentioned above, lncRNAs can also compete for protein binding to regulate the transcription or translation of a gene. However, such lncRNAs have not yet been reported in adipogenesis, in contrast to studies of cancer and neurodegenerative diseases, amongst others.

### 6.4. Involved in Histone Modification

H19 was the first lncRNA discovered by humans and contains five exons and four introns. The fifth exon contains an Rsal polymorphic restriction site, which is highly conserved in mammals. Previous studies show that H19 and microRNA jointly inhibit the differentiation of white adipocytes through histone modification. Recently, Schmidt et al. [88] found that monoallelic H19 activates brown fat thermogenesis in a cold environment. Inhibiting H19 expression in brown and white adipocytes using RNAi significantly decreased the expression of genes related to adipogenesis and differentiation of brown adipocytes. Furthermore, overexpression of H19 in cells promotes the oxidative metabolism of mitochondrial lipids without affecting glucose metabolism. Previous studies have shown that H19 is a single allele expressed only on maternal chromosomes. The overexpression of H19 in brown adipocytes inhibits the expression of alleles of paternal origin that promote white fat differentiation, promoting obesity resistance caused by diet and playing the role of a doorman [88]. H19 modifies chromatin histone by binding MDB1. This inhibits the expression of paternal alleles, maintaining the energy metabolism of brown fat, and participating in resisting obesity development.

Sun et al. [89] found that 175 lncRNA transcription was significantly dysregulated during adipogenic differentiation. It is worth mentioning that MIR31HG, located on chromosome 9 and miR-31, is the host gene of this transcript and is in the nucleus and cytoplasm of hASCs. In vivo experiments have shown that overexpression of MIR31HG promotes the differentiation of adipocytes or the accumulation of lipids. Numerous stable stem–loop structures in MIR31HG exist, with the necessary spatial conformation to interact with the chromatin-modifying protein. MIR31HG promotes histone methylation and acetylation of the FABP4, inducing the transcription of this gene and lipid production [90]. However, downregulating FABP4 expression inhibits the role of MIR31HG in adipogenesis. Similarly, ADINR and HoxA-AS3 promote the adipogenic differentiation of white fat by directly or indirectly inducing histone modification of the target genes [91,92].

### 6.5. Scaffold

Xu et al. [93] used RNA-seq to reconstruct mouse brown fat, inguinal white fat, and epididymal white fat transcriptome. He identified 1500 lncRNAs, including 127 BAT-restricted loci induced during differentiation. As a result, it was found that lnc-BATE1 is a targeted gene between C/EBPa, C/EBPβ, and PPARγ genes and produces polyadenylic acid transcripts spliced from two exons. lnc-BATE1 is overexpressed in brown fat. However, it was not detected in other tissues tested, and it is necessary to maintain the thermogenesis of brown fat in mice [93]. lnc-BATE1 weakens the inhibition of brown fat thermogenesis activity while suppressing the expression of white fat genes. In transaction, lnc-BATE1 selectively maintains the BAT gene expression, inhibits the expression of the WAT gene, and is a specific heterogeneous nuclear ribonucleoprotein (hnRNPU) required for the formation of sexually bound brown fat, localizing the subnuclear translocation of lncRNAs, and the formation of complexes required for brown fat secretion. Knockdown of lnc-BATE1 has little effect on lipid accumulation and cell morphology during differentiation but downregulates the expression of all brown fat markers, including Cidea, C/EBPβ, PGC1α, PRDM16, PPARα, and UCP1. In addition, inhibiting lnc-BATE1 affects the biological activities of mitochondria, demonstrated by the underexpression of mitochondrial genes and inhibited UCP1 protein expression. Cold stimuli upregulate lnc-BATE1 by 3–4-fold. In summary, lnc-BATE1 is at the heart of the brown fat formation process, the secretion of numerous mitochondrial proteins, including UCP1 and thermogenesis of brown fat cells, and the regulation of WAT browning. Furthermore, Bmncr can inhibit adipogenic differentiation by promoting the formation of TAZ and the RUNX2/PPARG transcription complex [94]. Previous studies have shown that LINC00473 is a marker of human thermogenic adipocytes [95,96].

Linc-ADAL, located between AQPEP and AP3S1 genes on chromosome 5, is a marker for adipocytes. The study demonstrated that linc-ADAL regulates adipocyte differentiation and lipogenesis by interacting with hnRNPU and insulin-like growth factor 2 mRNA-binding protein 2 (IGF2BP2) at distinct subcellular locations [97]. Recently, Li [98] reported that the ribonucleoprotein Blnc1/hnRNPU complex participates in the development and thermogenesis of brown and beige fat, mediated by Zbtb7b.

### 6.6. Regulate Target-Gene-Promoter Activity

Chen et al. [99] found that long-chain noncoding RNA lnc-U90926 is mainly located in the cytoplasm and expressed in white adipose tissue. Moreover, its expression decreases with the differentiation of 3T3-L1 preadipocytes and lipid accumulation. Additionally, the expression of lnc-U90926 is lower in the subcutaneous and visceral adipose tissue of obese mice than in nonobese mice. Mice models revealed that overexpression of lnc-U90926 reduced the differentiation of 3T3-L1 adipocytes, inhibited lipid accumulation, and reduced PPARγ2, FABP4, and AdipoQ mRNA levels, whereas lnc-U90926 knocking down exerted an opposite effect [99]. The double-luciferase assay revealed that lnc-U90926 inhibited the activation of the PPARγ2 promoter by binding at the −2000 bp~−1500 bp region, but this had no effect on C/EBPa activation. Since most lnc-U90926 is in the cytoplasm, while PPARγ2 is in the nucleus, it is unlikely that lnc-U90926 directly interacts with the PPARγ2 promoter. lnc-U90926 possibly inhibits PPARγ2 transcription by binding to its promoter. However, the specific mechanism needs further investigation. Similarly, Zhu [100] discovered Plnc1, a new lncRNA transcribed from the peroxisome proliferator-activated receptor γ2. Plnc1 increases the transcription of PPAR-γ2 by reducing the methylation of the CpG region in the PPAR-γ2 promoter region.

Maternally imprinted noncoding RNA Dio3os [101] promoter methylation in obese maternal oocytes increases the activation of Dio3 and reduces the effect of thyroxine T3. This inhibits the activity of genes related to brown fat thermogenesis, such as Prdm16.

### 6.7. Guide

Obesity-related lncRNAs (regulated lncRNAs in adipocytes of obese individuals) in the scapula, groin, and gonads have been identified. Among them, lnc-leptin is a 28 kb enhancer region upstream of leptin protein (leptin) and has two exons. The genomic site of exon 2 interacts with the Lep promoter, increasing the expression of the gene. The lnc-leptin transcription start site has a positive trimethylated H3K4 (H3K4Me3) signal and RNA-polymerase-II-binding site. Therefore, lnc-leptin is actively transcribed in adipose tissue. The expression is highest for eWAT, followed by iWAT and BAT. In tissues, lnc-leptin is underexpressed in the testes and eyes. lnc-leptin is mainly expressed in the adipose tissues. lnc-leptin mediates protein complex subunit 1 (MED1) promoting the general transcription of enhancer regions, gene promoters, and RNA polymerase II, the components of the complex. lnc-leptin promotes leptin protein expression by recruiting MED1 to the lnc-leptin and Lep promoter regions during the transcription process [102]. Fan et al. [103] found that slincRAD, a long noncoding RNA, promotes the translocation of DNMT1 protein to the nucleolus region during the S phase, essential for the epigenetic modification of genomic DNA. SlincRAD participates in DNA methylation by interacting with DNMT1 and promotes fat formation and motivates early cell expansion.

Paral1 is underexpressed in mature adipocytes, but its expression is upregulated during the adipogenesis. Paral1 knockdown in 3T3-L1 preadipocytes disrupts lipid formation, whereas the overexpression of Paral1 has no significant effect on the differentiation of adipocytes. Further functional validation experiments have shown that Paral1 activates the transcription of PPARγ and promotes the differentiation of preadipocytes by interacting with PSPC1 and hnRNP-like-binding protein 14 (RBM14) [104].

### 6.8. Cell Signal Pathways and Regulates Downstream Genes

#### 6.8.1. Cell Signal Pathways

There are many signaling pathways involved in adipogenesis, and there is a crosstalk between them, resulting in a complex regulatory network. The essence of the regulation of fat deposition is the regulation of adipocyte proliferation and differentiation. Therefore, the regulation of lipogenesis is associated with the activation of cell-proliferation-related signaling pathways. For example, the Phosphoinositide 3-Kinase/Protein Kinase B (PI3K/AKT) pathway regulates the stability of the cell-cycle-related gene cyclin D1 (CCND1) during adipocyte proliferation via AKT [105], indirectly promoting the differentiation of adiposity. Furthermore, Foxo1 interference can significantly inhibit adipocyte differentiation; thus, activation of the PI3K/AKT pathway positively correlates with lipogenic differentiation of cells [106]. As we know, phosphorylation of C/EBPβ transcriptional activation leads to its subsequent activation by PPARγ and C/EBPα transcription [107], promoting adipocyte entry into the terminal differentiation phase. MCE is a necessary stage for adipocytes to enter lipogenic terminal differentiation. Therefore, the MAPK/ERK-signaling pathway also has an active role in the process of lipogenic differentiation [108]. In contrast to these signaling pathways, the Wnt/β-catenin-signaling pathway plays a negative role in the regulation of lipogenic differentiation [109,110]. In the early stages of adipogenesis, C/EBPβ and C/EBPδ are activated in response to adipose-stimulating hormone and inhibit the Wnt-signaling pathway. It has been shown that the Wnt-signaling pathway is predominantly expressed in preadipocytes and has limited expression in mature adipocytes. Unphosphorylated β-catenin in the nucleus binds to TCF4 to regulate downstream target genes and represses C/EBPβ and C/EBPδ expression. AMPK, a central regulator of cellular energy sensors, has been reported to be involved in many biological processes such as cell proliferation, apoptosis, and carcinogenesis [111]. The TGF-β superfamily signaling pathway plays an important role in regulating the lipogenic differentiation of mesenchymal stem cells; unlike the above-mentioned signaling pathways, the TGF-β superfamily mainly regulates adipocyte commitment through SMAD and BMP ligands. The TGF-β superfamily is therefore considered to be a signaling pathway associated with the expression of white fat, brown fat, and beige fat marker genes.

lnc-ORA knockdown inhibits the transition of cells from the G1 to S phase. lnc-ORA participates in the proliferation and differentiation of preadipocytes via the PI3K/AKT/mTOR-signaling pathway by regulating DNA replication [112], involved in lipogenic differentiation of cells mainly by promoting phosphorylation of mTOR, AKT and PI3K. Additionally, lnc-OAD [113], lnc13728 [76], and lncRNA AC092834 [75] participate adipogenesis via regulating protein expression of β-catenin in the WNT/β-catenin-signaling pathway. RNA-seq analysis showed that lncRNA GM13133 promotes the browning of white fat by activating the intracellular cAMP transduction pathway and promoting the expression of brown fat marker genes such as PGC1α, UCP1, and BMP7, ultimately promoting the browning of white fat [114]. Similarly, lnc-uc.417 promotes the adipogenic differentiation of brown adipocytes by inhibiting the phosphorylation of p38/MAPK [115]. In the same way, lnc-FR332443 increases the expression of Runx1 in mouse adipocytes and inhibits adipocyte differentiation by attenuating the phosphorylation of MAPK-p38 and MAPK-ERK1/2 expression [116].

These lncRNAs are mainly involved in the lipogenesis process by regulating the expression of downstream genes through the phosphorylation level or protein level of key members of the pathway. Compared with other disease research areas, the number of lncRNAs regulating signaling pathways involved in lipogenesis is small, and researchers need to explore the relevant molecular mechanisms more deeply to provide research targets for the treatment of obesity and other metabolism-related diseases.

#### 6.8.2. Regulates Downstream Genes

SRA was the first lncRNA found to be related to adipocyte differentiation. The lncRNA is also called steroid receptor RNA activator (SRA) [117] and is encoded by the SRA1 gene. The expression level of SRA is twice that in preadipocytes relative to the mature mouse adipocytes, and it directly binds and promotes PPARγ transcription. In addition, SRA improves insulin resistance in mouse adipose tissue by inhibiting the phosphorylation of c-Jun NH2 terminal kinase induced by TNFα and inhibits the proliferation of adipocytes by promoting mitosis in these cells in the S phase [118]. In addition, mice models revealed that SRA could also regulate adipogenesis in bipotential ST2 mesenchymal cells by inhibiting the phosphorylation of p38/JNK in the early stages of adipogenesis and activating the expression of insulin receptor gene and downstream signal transduction [119]. NEAT1 [120] is mainly located in the nucleus. NEAT1 dysregulates the phosphorylation of Clk kinase by binding to the SR protein, affecting the variable splicing of PPARγ pre-mRNA and promoting the differentiation of adipocytes.

MALAT1 is a highly expressed lncRNA that regulates numerous physiological and pathological processes in many tissues, including myogenesis, cancer, aneurysms, etc. [121]. Han et al. [122] found that MALAT1 is mainly located in the nucleus, is expressed in fat cells, and is underexpressed in the subcutaneous white adipose tissue of cancer-related cachexia patients, which is related to the low-fat mass index and poor prognosis of cancer. Subsequent experiments showed that the expression of MALAT1 positively correlated with that of FABP4 and LPL genes that regulate, indicating that MALAT1 promotes fat formation [123]. MALAT1 regulates PPARγ gene expression and participates in adipogenesis at the transcriptional level via the PPAR-signaling pathway, fatty acid metabolism, and insulin signaling.

In addition, Ctcflos [124] operate by variable shearing and thus shearing out shorter Prdm16 isoform transcripts which participate in promoting white fat browning. Some lncRNAs promote or inhibit the expression of downstream genes such as adipogenic differentiation, but the molecular mechanism underlying the regulation of related downstream genes, such as PVT1, lncSAMM50, lncAK079912, lnc 2310069B03Rik, HOTAIR, Dreh, LIPE-AS1, and HoxA11-AS1 are not well understood [125,126,127,128,129,130,131,132,133].

## 7. The Role of LncRNA in Ectopic Fat Deposition

Ectopic fat is mainly deposited around organs with high metabolisms, such as muscle and liver [134]. Nonalcoholic fatty liver disease is caused by excessive deposition of ectopic fat in the liver. In severe cases, it can cause steatohepatitis and liver fibrosis, advanced liver disease, and liver cancer. Some lncRNAs regulate liver lipid deposition and can promote or inhibit liver lipid deposition through numerous mechanisms, such as the phosphorylation of the cell signal pathway members, cis-acting regulation of adjacent coding genes, and regulating the methylation of target genes [135,136,137]. Yang et al. revealed that lncLMS (liver metabolically sensitive lncRNAs) are dysregulated in the liver during refeeding syndrome. Among them, Gm16551 [138] is a relatively conserved lncRNA in mice and humans. Preliminary fluorescence quantitative analysis and biochemical index detection have shown that Gm16551 and lnc-HR1 [139] have a similar mechanism inhibiting the deposition of liver lipid deposits by blocking the activity of SREBP1c. Lan [140] found that lnc-HC, a new type of long noncoding RNA, regulates the expression of Cyp7a1 and Abca1 by modulating cholesterol metabolism in the liver. Additionally, it negatively regulates PPARγ at the translation and protein levels and inhibits the formation of lipid droplets in liver cells. In addition to regulating the growth and development of adipocytes, H19 improves insulin resistance by promoting fatty acid oxidation and reducing the accumulation of ectopic lipids in skeletal muscle by targeting hnRNPA1 [141].

## 8. LncRNA-Mediated Adipogenesis Dysregulation and Disease

There is no doubt that lncRNAs are an important source of potential targets against obesity and other related metabolic diseases. However, the molecular mechanisms underlying the cooperative regulation of lncRNAs with some nutrients and with classical signaling pathways are still unclear. In addition, most studies have used mouse 3T3-L1 preadipocytes or human-derived, adipose-derived mesenchymal stem cells as models for in vitro experiments, and many of the experimental effects of lncRNAs in vivo have not been validated. Therefore, it is important to investigate the regulatory mechanisms of lncRNAs and the regulation of lipogenic differentiation under pathophysiological conditions, or to explore the association between lncRNAs and certain disease models of lipid metabolism imbalance, which could better facilitate the development of lncRNA-targeted drugs and other related research.

Metabolism-related lipid disorders include atherosclerosis, obesity, Alzheimer’s disease, type II diabetes, tumors, and other diseases [142]. Numerous studies have shown that lncRNAs are involved in biological processes such as metabolic regulation, inflammation, immunity, and vascular function [89,143]. Unfortunately, the biological functions of most lncRNAs have not been fully annotated, and we can only use some cellular models and animal models to validate the functions of lncRNAs in diseases related to lipid metabolism disorders at this stage [144]. In addition, exploring the role of lncRNAs in diseases with abnormal lipid metabolism has some reference significance for the development of targeted drugs for related diseases.

Lipid metabolism in cancer stem cells is important for drug sensitivity and maintenance of stemness in cancer cells. Liu [145] et al. showed that lncROPM enhances the stability of PLA2G16 mRNA by directly binding to the 3’-UTR of PLA2G16, thereby activating active PI3K/AKT, Wnt/β-catenin, and Hippo/YAP signaling, ultimately involved in the maintenance of stemness and the enhancement in chemoresistance in BCSCs. In a mechanistically similar fashion, Hilnc promotes PPARγ expression through the formation of an RNA-RNA stable dimer with IGFBP2 and promotes lipid deposition in the liver [146]. In atherosclerosis, kcnq1ot1 enhances HDAC3 expression by competitively binding to miR-452-3p, thereby inhibiting ABCA1 expression as well as cholesterol efflux. kcnq1ot1 promotes macrophage lipid accumulation and accelerates the development of atherosclerosis via the miR-452-3p/HDAC3/ABCA1 pathway [147]. In addition, as mentioned above, lncRNA is also a good potential biomarker. lnc-P3134 was significantly upregulated in the blood exosomes of patients with type 2 diabetes [148] and, in addition, lncH19 was upregulated in the serum of patients [149]. lncRNAs have a very important impact on lipid metabolism homeostasis, but the specific regulatory mechanisms and biological processes such as lncRNA regulation of lipid production, β-oxidation, and lipid catabolism need to be further investigated. In-depth investigation of the regulatory functions of lncRNAs in lipid metabolism homeostasis has important biological implications for lncRNAs to become biomarkers for certain diseases.

## 9. Summary and Outlook

At present, researchers reveal the role of lncRNA in adipose tissue development, and provide new insight into treating metabolic syndrome and the best animal husbandry practices to produce the best animal products. Studies have shown that lncRNA plays an important role in regulating lipogenesis, thermogenesis, and lipid metabolism. However, compared with other human diseases, fewer lncRNAs are associated with obesity, adipose tissue development, and fat deposition. Research on fat-related lncRNA is still at infancy, and many lncRNA-regulatory mechanisms remain to be discovered. Investigating the regulatory patterns of lncRNAs in adipose-tissue-derived MSCs or progenitor cells could better explain their role in lipid deposition. Furthermore, the small individual differences in MSCs compared to mature adipocytes are more suitable for functional studies of lncRNAs at the cellular level. However, few studies on lncRNAs for lipogenic differentiation have been carried out in ADSCs at this stage. lncRNAs are a potential biomarker for numerous diseases. Additionally, lncRNA can be used to develop pharmacological compounds that regulate adipocyte differentiation and, thus, treat obese patients. However, lncRNA’s muscular and time–space specificities restrict fat-development-related lncRNA studies. The diverse mechanisms of action of lncRNA further complicate related research. Moreover, given the technical limitations, studies on the lncRNA structure remain scanty. The lncRNA function dramatically limits the discovery of the function.

Advances in molecular biology techniques and high-throughput sequencing can reveal more about lncRNA and their functions more accurately. For example, the pull-down assay revealed protein binding of lncRNA [150], whereas high-throughput sequencing of RNA reveals the expression level of lncRNA [151]. DNA microarray can quickly obtain the sequence and expression level of a large number of lncRNAs [152]; RNA-binding protein immunoprecipitation (RIP) can be used for studying the binding between lncRNA and proteins in cells [153]; the Western blot method revealed proteins regulated and bound by lncRNA [154]. Furthermore, bioinformatics analyses can reveal the function and mechanisms underlying the functioning of lncRNA. Databases such as UCSC, KEGG, and GSDB continue to expand lncRNA research. lncRNAs that participate in adipokinesis and fat deposition in the body have been reported. Due to lncRNAs having relatively stable secondary structure characteristics, marker lncRNAs for obesity and nonalcoholic fats diseases, such as liver and insulin resistance, can be identified. Translated nucleotides (ASOs) stably reduce the expression of target RNA. Therefore, for obesity treatment, specific lncRNAs that participate in obesity development can be targeted and regulated.

Since the discovery and naming of the first nucleic acid Nuclein in 1869, more than 270,000 lncRNAs, including more than 1800 lncRNAs of known functions, have since been discovered and recorded in the LncBook. Nucleic acids have been studied for over 151 years. The functions of the major biomolecules has gradually become known, but the understanding of lncRNA is still at infancy. Given the poor lncRNA conservation among many species, it is cumbersome to accurately locate homologous lncRNA. In addition, the regulatory mechanism of lncRNAs differs between cell sites. The mode of lncRNAs’ regulation of gene expression is regulated through complex mechanisms. For example, lncRNAs can also form RNA double-stranded inhibitory RNA, translate or enhance the stability of RNA, and some can also absorb miRNA like a “sponge”. Studies have also shown that most lncRNAs lack ORFs and do not code for proteins. However, about 40% of lncRNAs are translated into nonfunctional peptides, and a few lncRNAs can encode functional peptides with specific biological functions [155]. In addition, BAT significantly differs between different species and age groups of the same species.

For humans, BAT mainly exists in the perinephric, neck, supraclavicular, and pericardial regions. In rodents, BAT mainly gathers in the perinephric, scapular, mediastinum, axillary, and axillary areas at birth. The distribution of BAT in the neck area differs between species. In addition, there are very significant differences in the BAT content among species of different ages. In sheep, BAT accumulates in the perirenal adipose tissue at birth and then rapidly decreases after 30 days. ELD is caused by excessive lipid accumulation in other organs or tissues ectopically, damaging various organs of the body through lipotoxicity. Therefore, most research reports are related to pathological factors and reveal ways of regulating ELD due to the nonconservation of lncRNA and the differences in the expression of BAT in different tissues and different age groups. However, in recent years, it has been possible to understand the mechanism of action of lncRNA on fat based on the characteristics of the biomolecules. The research of transcription factors (TFs) related to fat development has deepened our understanding of fats, for example, its representative C/EBPa, C/EBPβ, PPARγ, and so on. There is a strong correlation between the expression of specific TFs and fat secretion. When studying the development of BAT and the browning of WAT, UCP1, as an indispensable uncoupling protein in the non-trembling thermogenesis of brown fat and beige fat, can also be used as a target for browning-related lncRNA research. Therefore, in recent years, research on fat-related lncRNA has aimed to quantitatively analyze the above-mentioned or other transcription factors or proteins related to fat development, such as the indirect UCP1 inhibitor, Cidea, and many other regulatory factors. These mechanisms further identify lncRNA that regulates fat secretion and the mechanism underlying this process.

lncRNA is poorly evolutionarily conserved, and the diversity in their regulatory mechanisms complicates related research. However, the mechanism of action of lncRNA follows certain principles. Although few lncRNAs can encode functional peptides, the number of non-protein-coding sequence lncRNAs tends to increase with the complexity of organisms. The increase in the number of lncRNAs with organism complexity provides a preliminary basis for a better analysis of the coding capabilities of lncRNAs. Studies have shown that the evolutionary conservation of lncRNA is low, which is thought to regulate the plasticity [155]. Comparing short sequences or structural elements rather than searching for functional equivalence through genome-wide alignment is a feasible approach. For example, by comparing short sequences such as binding sites for AREs, it is possible to locate homologous lncRNAs in different organisms. In addition, lncRNAs have higher tissue specificity than the protein-coding genes. Compared with protein-coding genes, lncRNAs have fewer exons, shorter lengths, and lower levels of protection and expression. Therefore, cytoplasmic lncRNAs move across the cell membrane by combining with specific proteins. Some lncRNAs communicate with and regulate the expression of protein-coding genes through direct competitive miRNA binding, thereby mimicking ceRNAs. Therefore, homologous lncRNAs can be identified using specific miRNAs. Biological functions, such as biosynthesis of unsaturated fatty acids involved in the biosynthesis of fat secretion; fatty acid elongation and metabolism; citric acid cycle; oxidative phosphorylation, mitochondrial respiratory chain Complex enzymes; and the AMP-dependent protein kinase (AMPK)-signaling pathway can also reveal related lncRNAs.

We would like to conclude that this section of lncRNA research related to fat secretion is to identify therapeutic targets for treating obesity and related syndromes and molecular markers for obesity diagnosis. This article reviews the long-chain noncoding RNAs related to fat development, aiming to classify the functions and mechanisms of lncRNAs to find the characteristics and commonalities of different lncRNA targets. It also summarizes some experiments and develops research techniques and methods for lncRNA related to fat development to seek more advanced research methods. Finally, the research on lncRNA related to fat development cannot stop here. Numerous lncRNAs, their function, and related mechanisms remain to be discovered.

## Figures and Tables

**Figure 1 ijms-23-07488-f001:**
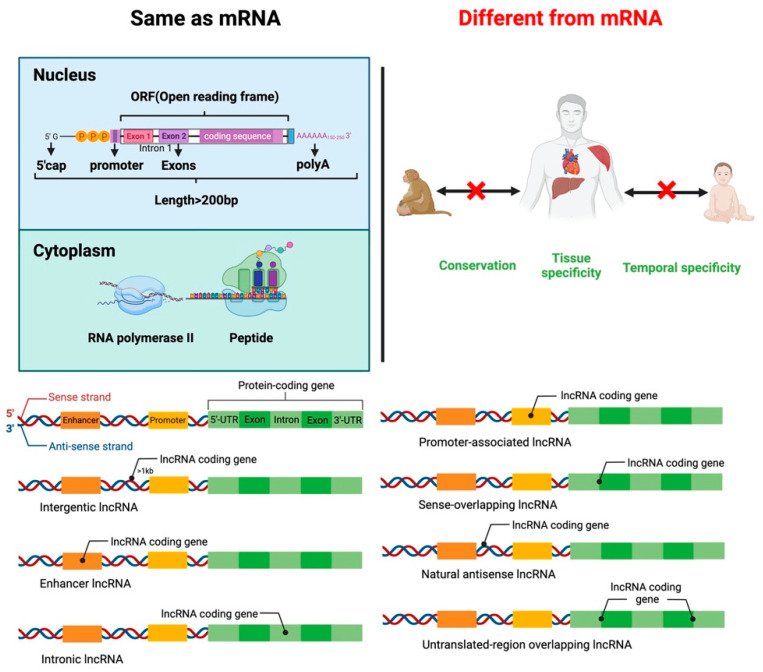
Similarities and differences between lncRNA and mRNA structure and function. Classification of lncRNAs according to the transcription position of the genome.

**Figure 2 ijms-23-07488-f002:**
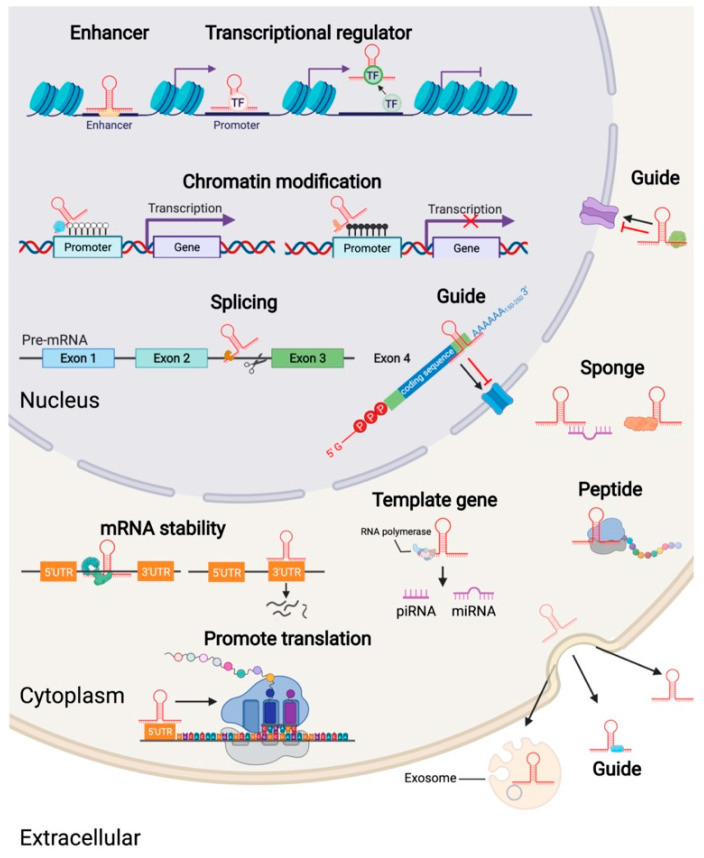
Biogenesis and function of long noncoding RNAs (lncRNAs). The lncRNA in the cell nucleus exerts its biological functions through the regulation of transcription, chromatin modification, variable splicing of mRNA, and changing the subcellular localization of mRNA. In the cytoplasm, lncRNA exerts biological functions by affecting the stability of mRNA, translation of mRNA, sponge-adsorbed protein, or miRNA. In addition, a small number of studies have shown that lncRNA can be transcribed to produce small peptides and can also function as precursor molecules of piRNA and miRNA. In addition, lncRNA can also be excreted outside the cell through exosomes, tricking proteins into transporting out of the cell or being excreted outside the cell alone to perform functions such as crosstalk.

**Figure 3 ijms-23-07488-f003:**
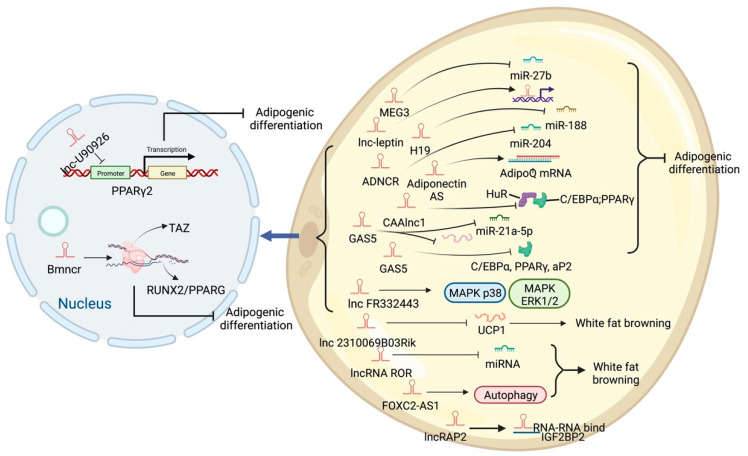
Schematic diagram of the biological function of lncRNA that promotes lipid deposition of white fat.

**Figure 4 ijms-23-07488-f004:**
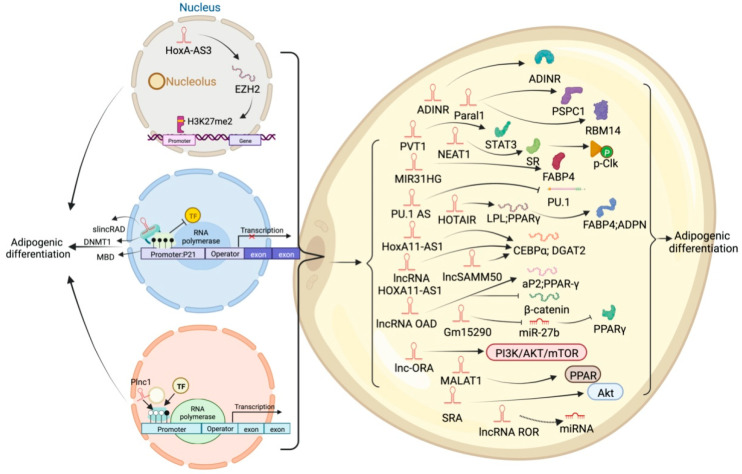
Schematic diagram of the biological function of lncRNA that inhibits or participates in the deposition of white fat lipids.

**Figure 5 ijms-23-07488-f005:**
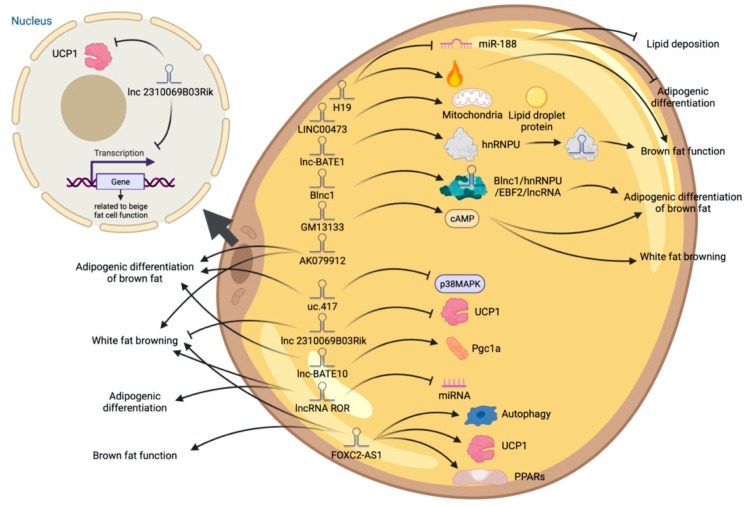
Schematic diagram of the biological function of lncRNA involved in the differentiation of brown adipocytes.

**Table 1 ijms-23-07488-t001:** Adipogenesis and development-related lncRNAs (miRNA Sponge).

LncRNA	Target miRNA	Species	Year
NEAT1	miR-140	Human	2015
Gm15290	miR-27b	Mouse	2017
MEG3	miR-140-5p	Human	2020
H19	miR-188	Mouse	2018
TINCR	miR-31-5p	Human	2018
GAS5	miR-18a	Human	2018
GAS5	miR-21a-5p	Mouse	2018
H19	miR-30a	Human	2019
lnc PGC1β-OT1	miR-148a-3p	Mouse	2019
LncRNA HCG11	miR-204-5p	Human	2020
lncRNA RP11-142A22.4	miR-587	Human	2020
lncRNA-Adi	miR-449a	Human	2020
lncNEF	miR-155	Human	2021

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
