# Peer review of "LncRNA-Mediated Adipogenesis in Different Adipocytes"

_ijms, 2022, doi:10.3390/ijms23137488_

Round 1

Reviewer 1 Report

The review article entitled “LncRNA-mediated adipogenesis in different adipocytes” by Zhang et al, comprehends the importance of a special type of non-coding RNAs – lncRNA and its involved in adipogenesis and related complexity. Authors have summarized the significance of lncRNAs via multitude of mechanism to regulate the post transcriptional processes which might be interesting to the readers of the journal.

I have few concerns about the manuscript and conclusions drawn by authors as below:

1)    In figure 1, the human diagram represents as if the conservation and temporal specificity is absent only in the highlighted tissue. I strongly believe this is not the case and suggest the authors to show human body with no tissue and let the tissue specificity of lncRNAs open for all.

2)  Authors have illustrated all the possible mechanism that involve lncRNAs except for an important aspect of "sponging" phenomenon in the context of RNA binding proteins. Authors need to include this mechanism in figure 2 and the text of their review article as well. They can refer to PMIDs: 31150052, 33441968, 31822653 etc.

 Minor:

a)       Line 32: 1%-2% should be 1-2%

b)      Line 41 and 42 require relevant references for authors’ statements.

c)    Line 75: lncRNA s(intergenic lncRNA) should be lncRNAs (intergenic lncRNA)

d)      Line 92: “..” should be “.”

e)    Line 225: I am not sure if is it Table 1 (missing) or Supplementary Table 1, please correct it accordingly.

f)     Line 565: In summary, the purpose of summarizing …. I think authors need to paraphrase this and such text throughout the manuscript.

g)   Bibliography require a quick revision as per the reference style and formatting style suggested by the journal. For instance – ref. 11, 12, 71, 75, 106, 112, 124, 125 are with doi and others are not.

h)  Authors are strongly encouraged to proof-read the manuscript for sentence composition and grammatical inconsistency to meet the formatting standard of the journal. 

Reviewer 2 Report

This review by Zhang, et al. gave readers a complete overview of how LcnRNAs regulate adipose tissue functions. The background of LncRNAs and adipose tissue has been introduced, and the functions of LncRNAs have been described in various adipose tissues, such as white and brown adipose tissues. Overall, this is an informative review that has scientific merits. However, several issues need to be addressed.

Major:

1. Since the review focused on adipogenesis, how LncRNA functions in adipose precursor cells/progenitors of white, beige, and brown needs to be discussed. However, the current manuscript focused on mature adipocytes.

2. I don't think the classification of adipose tissues into white, brown, and ectopic lipid deposition is accurate. The beige adipose tissue is neglected. Ectopic lipid deposition couldn't be considered as adipose tissue.

Minor:

1. The English writing needs to be greatly improved. Many errors and incomplete sentences were recognized, such as Line 68-69 is an incomplete sentence.

2. In the abstract, "unlike white fat, brown and beige fats are energy storage domains" is a wrong claim. 

Reviewer 3 Report

The review manuscript by Zhang et al described the role of different lncRNAs in mammalian adipogenesis in various adipocytes. Overall, the review structure looks good with updated literature and graphical representation. However, the manuscript needs critical improvements in writing and elaborate some of the sections to make the review more effective. The reviewer concerns over the review are as follows:

1) Some of the terminology used in the abstract needs to be corrected: for instance ‘maturation levels’ in a sentence started with ‘multiple levels..’. maturation of what? and ‘chromosome structure’ should be written as ‘chromatin structure’ (lines 18/19).

2) The title of section -3 needs to be rewritten. ‘Mechanism of lncRNA’ means? Is it mechanism action of lncRNA or mechanisms of lncRNA functionality? Write clearly

3) check lines 127/128. Correct ‘nuclear subnuclear structure’

4) elaborate abbreviations at their first usage. For instance, NF-kB, COVID-19/SARS-Cov, etc.

5) In line 225, authors cited ‘Table-1’ that was not provided in the main manuscript. Correct it to ‘Table-S1’.

6) Section-5 was provided with THREE figures and ONE table with just 100 words. Authors should attempt to describe the few critical lncRNAs in different adipose tissue functionalities.

7) Authors started with just first author last name for describing their papers. Lines 330 (Sun found..) and line-342 (Xu used RNA..), line 373 (Chen found..) needs to be added ‘et al.’ after author last name as the work was not done just by one author.

8) Elaborate and list down (may be table like-Table S1 with lncRNAs and their target miRNAs) some of the critical miRNAs sponged by lncRNAs for regulating adipocytes differentiation (ref 78-90). Authors will get the spectrum of miRNAs from this review itself instead of going over 12-13 papers cited in line 313.

9) In line 310, format ‘Pparv’ to ‘PPARγ’ in consistence with other text.

10) elaborate 6.8.1 section-discuss the role of different signaling pathways in ‘adipogenesis’ and how these pathways were regulated by lncRNAs.   

11) in line 421, write ‘GM13133’ to ‘lncRNA GM13133’

12) Lines 421-424, cite 114 and 115 separately each of those sentences.

13) Adding an additional section discussing role of lncRNA-mediated adipogenesis dysregulation linking to in some disease states.

14) Please check for the uniformity of the references (p-15 to page-19). Some have Doi and some do not. Check ref-36 (journal name repeated), 86 (no author names & repetition of journal names). Please follow IJMS references guidelines for formatting the style.

15) Please update the ‘Table-S1’ by adding ‘citations’ which authors were referring.

16) Does the list of lncRNAs listed in Table-1 are all from humans? Or mix of lncRNAs from mouse too? If so, separate and indicate separately by adding ‘organism’ column.

17) Please add ‘Authors contribution’ & ‘Abbreviations’ sections at the end of the body text.

Best wishes,

Round 2

Reviewer 2 Report

The revised version is improved. However, many typos and writing errors remain. For example, in the Abstract, "brown and beige fats releases energy as heat" should be "release"...